# Effects of Pesticide Intake on Gut Microbiota and Metabolites in Healthy Adults

**DOI:** 10.3390/ijerph20010213

**Published:** 2022-12-23

**Authors:** Jun Ueyama, Mai Hayashi, Masaaki Hirayama, Hiroshi Nishiwaki, Mikako Ito, Isao Saito, Yoshio Tsuboi, Tomohiko Isobe, Kinji Ohno

**Affiliations:** 1Department of Pathophysiological Laboratory Sciences, Field of Radiological and Medical Laboratory Sciences, Nagoya University Graduate School of Medicine, 1-1-20 Daiko-minami, Higashi-ku, Nagoya 461-8673, Japan; 2Division of Neurogenetics, Center for Neurological Diseases and Cancer, Nagoya University Graduate School of Medicine, 65 Tsurumai-cho, Showa-ku, Nagoya 466-8550, Japan; 3Department of Neurology, Fukuoka University, 7-45-1 Nanakuma, Jonan-ku, Fukuoka 814-0180, Japan; 4Health and Environmental Risk Division, National Institute for Environmental Studies, 16-2 Onogawa, Tsukuba 305-8506, Japan

**Keywords:** pesticide exposure, intestinal environment, fecal short-chain fatty acid

## Abstract

Animal experiments have indicated that pesticides may affect gut microbiota, which is responsible for the production of short-chain fatty acids (SCFAs) and polyamines. Here, we present a preliminary observation of the relationship between pesticide exposure and fecal SCFAs and polyamines in Japanese adults. In total, 38 healthy adults aged 69 ± 10 years (mean ± SD) were recruited and subjected to stool and spot urine tests. Urinary dialkylphosphates (DAP), 3-phenoxybenzoic acid, and glyphosate were assayed as pesticide exposure markers of organophosphorus insecticide (OP), a pyrethroid insecticide, and glyphosate, respectively. Significant negative correlations (*p* < 0.05, Spearman’s rank correlation coefficient) were found between urinary DAP, fecal acetate (r = −0.345), and lactate (r = −0.391). Multiple regression analyses revealed that urinary DAP was a significant explanatory variable of fecal acetate concentration (*p* < 0.001, β = −24.0, SE = 4.9, t = −4.9) with some vegetable intake (adjusted *R*-square = 0.751). These findings suggest that OP exposure is independently associated with lower fecal acetate levels, which may contribute to human health in middle-aged and older adult groups. Given that the human gut environment has long-term effects on the host, studies on wide-range age groups, including children, are necessary.

## 1. Introduction

There are many different types of pesticides, such as insecticides, herbicides, and fungicides/bactericides, each designed to be effective against specific targets. The term pesticide refers to a group of chemical compounds used to control pests, including insects, rodents, fungi, and unwanted plants (weeds). Pesticides play a significant role in increasing crop productivity and have many public health applications. The amount of pesticides used in agriculture at the global level was 4.15 million tons in 2018 [1]. However, there is a wide disparity in pesticide application rates between different geographic regions. China, Japan, and the Republic of Korea have application rates above 10 kg/ha, while the world’s average pesticide use per hectare is 2.6 kg/ha [1].

To protect human health from the adverse effects of pesticides, the WHO and other government agencies in various countries have developed laws and threshold values regarding the use and residues of pesticides, such as pesticide maximum residue limits in food and occupational exposure limits. However, the presence of pesticides in food is a public concern, and these substances tend to persist in the environment. The potential relationship between organophosphorus insecticide (OP) exposure and health effects, such as poorer intellectual development [2], behavioral deficits [3,4,5], asthma [6], obesity [7], and decreased lung function [8], even in children, have been suggested in previous reports. However, little is known about the mechanisms by which OP exposure can lead to these effects.

Recent studies have established the importance of gut microbiota in human health, and many publications have highlighted its role as an important component of human physiology. The relationship between the intestinal environment and the development of various diseases, such as allergic diseases [9], intestinal diseases [10], obesity [11], neurodegenerative diseases [12], and mental illness [13] is being elucidated and is attracting much attention. Therefore, a better understanding of the factors that affect gut microbiota in early life through aging could provide insights into their effects on human health. The composition of gut microbiota, an intestinal environmental indicator, can reportedly be altered by various factors such as age [14], diet [15], administration of chemicals, antibacterial compounds [16], and environmental chemicals such as pesticides. Animal experiments have reported that pesticides such as OP chlorpyrifos [17], diazinon [18], pyrethroid insecticide permethrin [19], and herbicide glyphosate [20] can affect gut microbiota. Specifically, short-chain fatty acids (SCFA) and polyamine-producing intestinal microbiota are significantly altered by pesticide exposure. SCFAs modulate key host functions including nutrient processing, maintenance of energy homeostasis, and immune system development [21], and polyamines play a key role in stress resistance and cell proliferation [22].

According to these previous experimental findings, OP, pyrethroid, and glyphosate exposure may affect the intestinal environment in humans. However, limited data are available on the relationship between the daily exposure levels of these pesticide and fecal metabolite concentrations. Here, we present preliminary observations of the urinary concentrations of pesticide exposure markers, fecal SCFAs, and polyamines in healthy Japanese middle-aged and older adults who are exposed to higher OP concentrations as compared with young adults [23].

## 2. Materials and Methods

### 2.1. Study Subjects

All participants were recruited as a control group for a clinical observational study [12]. Written informed consent was obtained from all participants. A total of 66 healthy controls were recruited from two hospitals (Nagoya University Hospital and Fukuoka University Hospital) between September 2015 and July 2019. Only those who submitted stool and spot urine samples as biological samples were included in this study (*n* = 38). Self-administered questionnaires were distributed to the participants. The questions were designed to collect data on lifestyle habits that are primarily associated with gut microbiota composition, such as smoking status, alcohol consumption, and diet.

Urine and fecal samples were collected by the participants in their homes and delivered to the hospitals at 4 °C for urine and 0 °C for feces within 24 h. These samples were stored at −80 °C until analysis.

All studies were approved by the Ethical Review Committees of Nagoya University Graduate School of Medicine (approval no. 2016-0151) and Fukuoka University School of Medicine (2016M027).

### 2.2. Determination of Pesticide Exposure Biomarkers in Urine

Quantitative determination was performed for the following biomarkers of pesticide exposure: dimethylphosphate (DMP), dimethylthiophosphate (DMTP), diethylphosphate (DEP), diethylthiophosphate (DETP) for OP insecticide, 3-phenoxybenzoic acid (3PBA) for PYR insecticide, and glyphosate for glyphosate herbicide. These substances have been measured as a biomarker of each pesticide for assessing not only occupational but also low-dose environmental exposure.

The concentrations of DMP, DMTP, DEP, and DETP were measured using a previously established method [24]. Briefly, 1 mL of urine was pipetted into a test tube, and 1 mL of distilled water, 20 μL of formic acid, and 20 μL of internal standard (30 mg/L DMP sodium salt-d_6_, 30 mg/L DMTP potassium salt-d_6_, 5 mg/L DEP ammonium salt-d_10_, and 5 mg/L DETP potassium salt-d_10_) were added. After gentle shaking, the test tube was incubated at 37 °C in a water bath for 10 min to decrease urinary turbidity. The sample was then loaded onto an SPE column (Oasis WAX SPE column; Waters Corporation, Milford, MA, USA). The passed and eluate samples were injected into a liquid chromatography tandem mass spectrometry (LC-MS/MS) system, which was composed of an Agilent 1200 infinity LC coupled with an Agilent 6430 Triple Quadrupole spectrometer (Agilent Technologies, Santa Clara, CA, USA). The total concentration of four dialkylphosphates (DAP) (ƩDAP; the sum of DMP, DMTP, DEP, and DETP concentrations), two dimethyl forms (DMAP; the sum of DMP and DMTP concentrations), and two diethyl forms (DEAP; the sum of DEP and DETP concentrations) were used for statistical analysis as summarized OP exposure biomarkers. The detection limits (LOD) were 0.7, 0.8, 1.0, and 0.1 µg/L for DMP, DMTP, DEP, and DETP, respectively. The intra-day precisions were less than 10.7% relative standard deviation (% RSD).

Concentrations of 3PBA in urine were measured using a gas chromatograph coupled with a mass spectrometer (GC-MS) equipped with an electron ionization system, according to a previously established method [25] with some modifications. Briefly, a 2-mL urine sample, 20 μL of internal standard solution (1 mg 2-PBA/L of acetonitrile), and 0.5 mL HCl (6 mol/L) were mixed in a glass test tube. After incubation at 106 °C, the target metabolites were extracted with *tert*-butyl-methyl-ether. The extract was evaporated at 40 °C (heat block) until dry using a gentle nitrogen stream. The residue was dissolved in 250 μL of acetonitrile. After derivatization with 1,1,1,3,3,3-hexafluoroisopropanol and *N*-diisopropylcarbodiimide, the derivatized target metabolites were analyzed using GC-MS (Agilent 7890A equipped with a 5975C inert MSD System; Agilent Technologies). The detection limit for 3PBA was 0.04 μg/L, and the intra-day precisions were 8% RSD.

Glyphosate concentrations in urine were measured according to a previously established method [26]. The detection limit for glyphosate was 0.1 μg/L, and the intra-day precisions were 11% RSD.

Urinary creatinine concentrations were quantified using high-performance liquid chromatography (HPLC) with a UV detector, according to the methodology described by Ueyama et al. [27]. The coefficient of variation of intra-day precisions was 0.2%.

### 2.3. Metabolite Analysis in Fecal Samples

All fecal samples were collected in screw-top fecal containers (height: 54 mm, diameter: 28 mm, polypropylene) and brown screw caps with an integrated spoon (high-density polyethylene; SARSTEDT Inc., Nümbrecht, Germany) by the study participants at home. The samples were transported to the laboratory at 0 °C and stored at −80 °C. Freezing was performed using a freeze dryer (FDU-2110) connected to a drying chamber (DRC-1100, EYELA, Tokyo, Japan) [28]. After drying, the fecal samples were moved from the fecal containers to a disposable grinding chamber (MT 40, IKA, Staufen, Germany). The samples were then ground (IKA tube mill control), and the fine-grain dried fecal samples were stored at −30 °C.

Acetate, propionate, butyrate, and valerate levels were determined according to the procedure reported by Ueyama et al. [28]. Briefly, 20 mg of freeze-dried feces (FD) were mixed with 1000 μL of 5 mmol/L NaOH and 300 μL of H_2_O. The mixture was then shaken and centrifuged. The supernatant (333 μL) was mixed with 200 μL of H_2_O, 50 μL of hexanoic-6,6,6-d3 acid solution (internal standard), 200 μL of 2-methyl-1-propanol, 133 μL of pyridine, and 67 μL of isobutyl chloroformate. The resulting solution was then mixed for 1 min. Subsequently, the mixture was added to 0.3 mL of hexane, shaken vigorously for 10 min, and centrifuged. The organic phase (upper layer) was then transferred to a GC glass vial. Quantitative analysis was performed using an Agilent 7890A GC equipped with an Agilent 5975 inert mass spectrometer (Agilent Technologies).

Succinate and lactate levels were analyzed using a method different from that described above. Briefly, 20 mg of FD, 400 μL of H_2_O, 50 μL of urine, 10 μL of sulfosalicylic acid solution (1 mg/μL), and an internal standard solution (hexanoic-6,6,6-d3 acid) were mixed and centrifuged. The supernatant was then transferred into a new test tube and mixed with HCl and diethyl ether. After shaking and centrifugation, the upper layer was transferred into a new glass test tube. This was followed by drying using nitrogen gas at 40 °C. Derivatization was performed with *N*-tert-butyldimethylsilyl imidazole for 30 min at 60 °C. Finally, the solution was analyzed using GC-MS. The amounts of SCFAs, lactate, and succinate were represented as mg/g of FD weight.

Polyamines putrescine (PUT) and spermidine (SPD) concentrations in each fecal sample were measured using the protocol proposed by Xiong et al. [29]. After derivatization using *N*-(9-fluorenylmeth-oxycarbonyloxy) succinimide, polyamines were analyzed using LC-MS/MS, which was composed of an Agilent 1200 infinity LC coupled with an Agilent 6430 Triple Quadrupole LC/MS System (Agilent Technologies). 1,6-diaminohexane was used as an internal standard.

Urinary creatinine concentrations were measured using HPLC with a UV detector, according to the methodology described by Ueyama et al. [27]. Standard creatinine solutions were prepared in HPLC vials at concentrations of 2, 4, and 6 mg/dL with H_2_O after diluting the urine samples 20 times with water. Analyses were performed using an Agilent HPLC 1100 series system (Agilent Technologies).

The coefficient of variation for the intra-fecal variability of feces (calculated from six analytical measurements) was 9.3% for acetate, 8.7% for propionate, 9.4% for butyrate, 8.3% for valerate, 12.7% for lactate, 11.9% for succinic acid, 8.0% for PUT, 14.8% for SPD in feces, and 0.24% for creatinine in urine.

### 2.4. Sample Collection, DNA Isolation, and V3–V4 16S rRNA Sequencing

A QIAamp PowerFecal DNA Kit (QIAGEN, Hilden, Germany) was used to extract DNA from 20 mg of FD. To assure efficient bacterial DNA extraction, we used FastPrep-24 5G (MP Biomedicals, Qbiogene, Montreal, QC, Canada), rather than vortex mixing. Samples were homogenized using FastPrep-24 5G for three cycles at 6.0 m/s for 60 s in Solution C1 of the QIAamp PowerFecal DNA Kit with a mixture of beads (Lysing Matrix E, MP Biomedicals, Santa Ana, CA, USA).

The V3–V4 hypervariable region of the bacterial 16S rRNA gene was amplified using a pair of primers (341F, 5′-CCTACGGGNGGCWGCAG-3′ and 805R, 5′-GACTACHVGGGTATCTAATCC-3′). The KAPA HiFi HotStart Ready-mix PCR kit (KAPA BIOSYSTEMS, Wilmington, MA, USA) and Nextera XT index kit (Illumina, San Diego, CA, USA) were used to prepare a metagenomic sequencing library. Using the MiSeq System (Illumina), nucleotide fragments were sequenced using the MiSeq Reagent Kit V3 (600-cycle). QIIME2 was used for taxonomic analysis. FASTQ files were quality-filtered, and DADA2 was used to generate amplicon sequence variants. The samples were not discarded during any of the filtration steps. For taxonomic analysis, we used the pre-trained naive Bayes classifier and the q2-feature-classifier plugin from QIIME2. The q2-feature-classifier was used to produce a trained reference from the SILVA taxonomy database and released 138 for taxonomic identification.

We filtered the taxa at the genus and family levels under the following conditions: taxa of which the average relative abundances were greater than 0.01% were set as analysis objects. Therefore, 64 families and 206 genera were selected.

### 2.5. Statistical Analysis

Kolmogorov–Smirnov tests were used to examine deviations from the normal distribution of the data, and these tests indicated that no urinary metabolites of pesticides had a normal distribution. Therefore, non-parametric tests were used for all statistical analyses. A comparison of fecal metabolites between the low- and high-pesticide exposure groups was performed using the Mann–Whitney U test. Forward stepwise regression analysis was performed using fecal acetate and lactate as the dependent variables. In model 1, age, sex, and body mass index (BMI) were used as independent variables. In addition to these clinical variables, food intake variables (intake frequency, times/week), supplement intake (yes/no), drinking habits (beer cans/day, other drinking times/week), smoking habits (never/past/current), delivery situation (natural/cesarean), and participant’s method of feeding as a newborn (breast milk/baby formula) were used as independent variables in model 2. The model with the lowest Akaike information criterion (AIC) was deemed to best fit the data. All statistical analyses were conducted using the statistical software JMP^®^ 15 (SAS Institute Inc., Cary, NC, USA), and two-sided *p*-values < 0.05 were considered statistically significant.

To determine which taxa were differentially represented between the low- and high-exposure groups, we performed differential abundance analysis by analyzing the composition of microbiomes with bias correction (ANCOM) and Wilcoxon rank-sum tests at the phylum, class, order, family, and genus levels [30,31].

## 3. Results

### 3.1. Study Subjects and Metabolite Analyses

The characteristics of the 36 subjects who were included in the final analysis are shown in Table 1. Overall, the mean age was 68.7 years (range: 39–81 years). Among them, there were six subjects (17%) with obesity (BMI > 25), three current smokers (8%), more than 28 subjects (78%) who underwent natural birth, more than 27 subjects (75%) who breastfed when they were infants, and four subjects (11%) with low stool frequency (3 times/week or less). Although several physical changes and health issues are more common with increasing age, information about any basic diseases was not collected in this study. Moreover, no information was collected on the participants’ occupations, such as agriculture.

The concentrations of SCFAs, polyamines, and pH in the feces and pesticide metabolites in the urine are summarized in Table 2. Non-parametric distributions were observed for all metabolite concentrations. These values are difficult to compare with other values because of the lack of reference values (or reference intervals) in clinical settings and exposure limit values in occupational health settings.

### 3.2. Correlations of Pesticide Exposure Markers and Fecal Metabolites

As shown in Table 3, significant negative correlations were found between lactate, SPD, acetate, and some insecticide exposure markers. The highest correlation coefficient was observed between lactate and DMAP, and the second lowest was observed between lactate and 3PBA. Significant correlations were also observed between acetate, lactate, and DAP. There was no correlation between the herbicide glyphosate and fecal metabolites. Outlier box plots of the concentrations of fecal SCFAs, polyamines, and pH in the high- (red) and low- (blue) pesticide exposure groups (cut-off value: median) are shown in Figure 1. Outliers were defined as points outside the whiskers, which were drawn to the furthest point within 1.5× the IQR (the 3th quartile minus the 1st quartile) from the box. Comparisons between the two groups were performed using the Mann–Whitney U test. The median concentrations of fecal acetate and lactate in the high-DAP groups (10.2 and 0.41 mg/g FD, respectively) were significantly lower than those in the low-DAP groups (17.3 and 0.76 mg/g FD, respectively). Moreover, the same concentration differences for acetate and lactate were observed in the DMAP group (data not shown). These results were not observed for the low- and high-DEAP, 3PBA, and glyphosate groups. These results suggest that exposure to OP may be associated with gut microbiota metabolites, such as acetate and lactate.

In the stepwise regression analysis, in which acetate and lactate were adopted as the criterion variables, DAP, DMAP, age, and BMI were detected as explanatory variables with low *R*-square values in model 1 (Table 4). Moreover, pesticide metabolite, food intake frequency, and supplement intake were also detected as explanatory variables with high *R*-square values of 0.751 for acetate and 0.920 for lactate in model 2 (Table 5). In model 2, only the OP metabolite DAP in urine was detected as an explanatory variable for acetate, but not for lactate. As shown in the Appendix A, there was no significant association between urinary DAP concentration and vegetable intake frequency, possibly the main OP exposure source [32,33], and other food intake frequencies. Furthermore, the concentration of urinary DAP was not significantly correlated with the food intake frequency of pasta, potatoes, fermented milk, and beans (data not shown).

### 3.3. Effect of OP Exposure on the Composition of the Gut Microbiome and Community Diversity

Based on the results of the stepwise regression analysis, we focused on identifying the relationship between urinary DAP levels (OP exposure markers) and gut microbial taxonomies and diversity. The genus level with composition differentiation between the low- and high-OP exposure groups with a *p*-value of 0.05 or less (Wilcoxon rank sum test) is listed in the Appendix A. Specifically, using ANCOM at the genus level, *Agathobacter* (W = 111) was found to be differentially abundant between the low- and high-OP exposure groups with a detection level of 0.7. However, none of the bacteria showed an FDR (Wilcoxon rank sum test) of 0.05 or less, indicating that gut microbiota composition did not appear to be affected by OP exposure. Additionally, no statistically significant difference in microbial alpha diversity (Shannon’s diversity index) was observed between the low- and high-DAP groups (*p* = 0.31; Mann–Whitney U test). Moreover, alpha diversity was not associated with urinary DAP concentrations (Spearman’s rank correlation coefficient test, r = 0.22, *p* = 0.20).

## 4. Discussion

This study for the first time utilized human biomonitoring of urinary pesticide exposure markers, the gut microbiome, and the quantitative metabolite profile to explore the influence of pesticide exposure on the intestinal environment. As the proposed approach could be applied successfully to research on adults, we believe it can also be employed in a large-scale study on children. The main finding of this study was that OP exposure is a potential risk factor for reducing fecal acetate in the human body. The results of some of our statistical analyses also suggest that lactic acid and SPD levels in feces may be correlated with insecticide exposure. However, the underlying mechanisms remain unknown.

Decreases in *Firmicutes*, *Bacteroidetes*, and *Lactobacillaceae* have been observed in mice following OP chlorpyrifos administration [34], and a decrease in Bacteroides has also been observed in rats [17]. Gao et al. [18] reported that OP diazinon affects the gut microbiome in a sex-specific manner. In females, the abundance of several families such as *Lachnospiraceae*, *Ruminococcaceae*, *Clostridiaceae*, and *Erysipelotrichaceae* decreased after exposure to diazinon. Some of these fecal microbiomes are involved in the production of SCFA. However, a significant alteration in gut microbiome composition caused by OP exposure was not observed in our study. This discrepancy might be related, in part, to different OP exposure duration and exposure levels as well as species differences between humans and experimental animals. Although the administered dose of pesticides in the aforementioned animal experiments was over 1 mg/kg/day, the daily intake of typical OP-compound diazinon in Japanese populations has been estimated at 0.02 mg/human/day [35]. We previously reported the concentration of pesticide exposure markers in urine obtained from a wide range of populations [26,28,36]. According to these data, the present study subjects did not have an extremely high level of pesticide exposure markers in their urine, indicating that no participants had been exposed to a high dose of pesticides occupationally or accidentally. Given that some OPs can alter microbial activities [37] and enzyme activities related to fatty acid metabolism [38], we also considered the possibility of the disruption of the biosynthesis of SCFA acetate through acetyl-CoA and the Wood–Ljundahl pathway, which is related to acetate production [39], as one of the reasons for the low concentration of fecal acetate in the high-OP exposure group. However, no evidence directly supported this hypothesis.

A wide range of factors might contribute to human exposure to OP pesticides, with vegetable or fruit consumption playing a significant role, as reported in some studies [40,41,42]. Therefore, the intake amount of vegetable and fruit dietary fiber, which is converted to monosaccharides and end-product SCFAs [43], might underlie the association between OP intake and low fecal acetate. However, no association was found between any food intake information and urinary OP exposure markers in this study. This result may suggest that the OP exposure decreased in a gut SCFA concentration, independent of dietary nutrient intake. Vegetable and other food intake information obtained from a brief survey could not explain the OP exposure level in this study, suggesting that OP exposure may have occurred via various food sources and not from a single one. Further investigation with questionnaire data that could potentially identify contributing factors of fecal SCFA concentrations, such as data obtained from a more detailed Food Frequency Questionnaire (FFQ) [44,45,46] compared to the simple FFQ used in this study, is needed to elucidate the relationship between fecal metabolites and OP exposure.

Acetate, the most abundant SCFA derived in the gut, binds to the G-protein coupled receptors GPR43 and GPR41 after absorption [47,48]. Animal and human data demonstrate that acetate beneficially affects a leaky gut by immunomodulating pro-inflammatory mechanisms, body weight control by appetite regulation, glucose homeostasis, and, potentially, insulin sensitivity [48]. There are still inconsistencies and differences between experiments that require further study. However, investigations should continue to determine whether exposure to pesticides and/or various chemicals is a variable factor in gut SCFAs and other metabolites.

Polyamines induce maintenance of the intestinal mucosal layer [49] and suppress inflammation by inhibiting macrophage inflammatory cytokine synthesis [50]. Commonly, polyamines in the intestine are indispensable substances for healthy mammals. Fecal polyamines are mainly produced by the intestinal microbiome, such as by *Enterococcus faecalis* [51]. In this study, no associations were found between urinary pesticide exposure markers and polyamines. Further research should be conducted for a better understanding of the potential association between pesticide exposure and fecal polyamines, which have a substantially increased diversity as compared with that of SCFAs.

More recently, Mesnage et al. [52] reported a relationship between dietary exposure to 186 common insecticides, herbicides, or fungicide residues and the fecal microbiome in 65 adult twin pairs in the UK. They provide evidence of an association between pesticide excretion and changes in gut microbiome metabolism. However, they did not mention whether fecal SCFAs and polyamines were included in their non-target metabolomics results. Our study, which features a quantitative analysis of fecal metabolites that are closely linked to health, strongly suggests the need for future cohort studies to understand the impact of pesticide exposure on gut environments and related health implications. A birth cohort study is particularly important because the human gut environment in early life has long-term effects on the host, and its aberrance may affect health in adulthood.

Multiple factors influence the composition of the human gut microbiome. Along with clinical outcomes such as allergic [11] and intestinal diseases [12], diet, drugs, and anthropometric measurements have been reported to explain over 20% of inter-person microbiome variability [53]. Odamaki et al. [54] has reported age-related changes in the gut microbiota composition of Japanese newborns to centenarians. Age-related sequential changes in gut microbiota at the phylum/genus level were found, but a similar composition was observed for people aged 20 to 60. The relative abundance of *Bacteroides* (phylum) and alpha-diversities of gut microbiota increased in people older than 70 years. The mean age of our study subjects was approximately 70 years. It remains unclear, however, whether age-related changes in gut microbiota composition affected our observations. Further study is needed to clarify the relationship between the gut environment and pesticide exposure in other age groups, including children.

Our study had some limitations. First, we were not able to obtain conclusive results because of the small sample size, indicating that representative sample collection was not performed in this study. However, as shown in Table 2, the fecal SCFA and urinary OP metabolite concentrations were similar to previous reports [27,28,55], indicating that our study subjects may not have significantly differed from other Japanese subjects. Secondly, the decrease in the absolute amount of bacteria producing SCFAs was not examined in this study. Nasuti et al. [19] reported that exposure to insecticides negatively affected gut microbiota in rats, with a decrease in the number of some SCFA-producing strains such as *Prevotella*. Thirdly, single-spot urine is not the most accurate sample to measure pesticide exposure chronically. Future studies should be conducted using 24-h urine or multiple-spot urine samples to better estimate pesticide exposure levels.

## 5. Conclusions

Our study explored the relationship between pesticide exposure markers and fecal metabolites, and identified a negative association between OP exposure markers and fecal metabolites such as acetate and lactate. Our results suggested that pesticide exposure in the middle-aged and older adult group could be negatively correlated with gut environment. This relationship can be used to better understand how pesticides, such as OP, may contribute to the health of this age group. Given that the human gut environment has long-term effects on the host, studies on a wider age range, including children and pregnant women, are necessary. Moreover, future studies are needed to confirm our observations.

## Figures and Tables

**Figure 1 ijerph-20-00213-f001:**
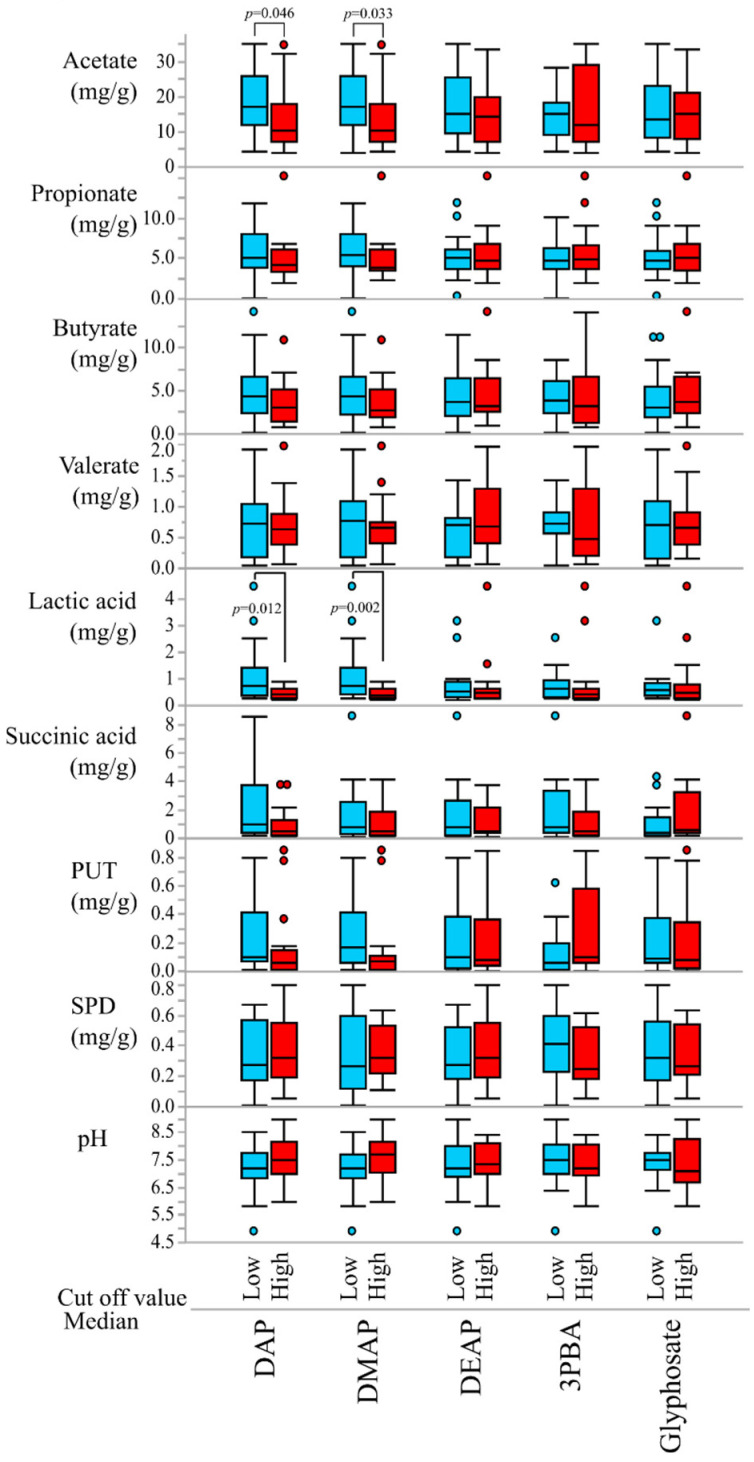
Outlier box plots of the concentrations of fecal SCFA, polyamine, and pH in low- and high-pesticide exposure groups (marker cut-off value: median). Whiskers are drawn to the furthest point within 1.5× the interquartile range (IQR) of the box. A comparison between the two groups was performed using the Mann–Whitney U test.

**Table 1 ijerph-20-00213-t001:** Demographic characteristics of the participants in this study.

Parameters		
Subject [*n*] (male/female)		36 (18/18)
Age [years]	Mean ± SD (range)	68.7 ± 9.9 (39–81)
BMI [%]	Mean ± SD	22.6 ± 3.0
Smoking [*n*]	Never	19
	Ex	7
	Currently	2
Natural birth [*n*]	Yes, No, No data	28, 0, 8
Breast milk [*n*]	Yes, No, No data	27, 1, 8
Stool frequency (/week)	Mean ± SD (range) median	9.0 ± 5.6 (2–21) 7.0
Fecal water content [%]	Mean ± SD (range)	74 ± 7 (55–86)

**Table 2 ijerph-20-00213-t002:** Concentration of pesticide exposure markers in urine and fecal metabolites of SCFAs and polyamines.

	Detection Rate (%)	Concentrations		
		Median	Mean ± SD	Range
Fecal metabolites				
Acetate (mg/g)	100	15.1	16.1 ± 9.2	3.9–35.1
Propionate (mg/g)	100	4.9	5.4 ± 3.0	0.1–15.2
Butyrate (mg/g)	100	3.4	4.4 ± 3.2	0.1–14.2
Valeric Acid (mg/g)	97	0.69	0.73 ± 0.50	<LOD–1.97
Lactate (mg/g)	100	0.46	0.80 ± 0.94	0.22–4.72
Succinic acid (mg/g)	100	0.52	1.43 ± 1.87	0.06–8.62
Putrescine (μg/g)	100	90	210 ± 262	5–846
Spermidine (μg/g)	100	281	352 ± 213	12–806
Pesticide exposure markers				
DMAP (μmol/g cre)	100	0.18	0.43 ± 0.94	0.07–5.76
DEAP (μmol/g cre)	100	0.06	0.18 ± 0.37	0.01–2.02
DAP (μmol/g cre)	100	0.25	0.61 ± 1.02	0.09–5.91
3PBA (μg/g cre)	97	0.59	1.01 ± 2.25	<LOD–14.02
Glyphosate (μg/g cre)	64	0.15	0.13 ± 0.14	<LOD–0.43
pH		7.2	7.4 ± 0.9	4.9–9.0

**Table 3 ijerph-20-00213-t003:** Significant correlations (Spearman’s rank correlation coefficient) between fecal metabolites and urinary pesticide markers at each concentration.

Fecal Metabolite	Urinary Pesticide Markers	Correlation Coefficient	*p*-Value
Lactate	DMAP	−0.571	0.005
Lactate	3PBA	−0.477	0.022
Lactate	DAP	−0.391	0.024
SPD	3PBA	−0.367	0.036
Acetate	DAP	−0.345	0.043

**Table 4 ijerph-20-00213-t004:** Stepwise regression analysis model with acetate and lactate as the criterion variables (Model 1).

	Acetate	Lactate
*R*-square	0.099	0.236
Predictors	β, SE, t, *p*	β, SE, t, *p*
Constant Age Sex BMI Urinary pesticide markers DAP DMAP DEAP 3PBA Glyphosate	18.96, 2.11, 8.98, <0.001 −6.23, 3.27, −1.91, 0.066	0.77, 1.44, 0.53, 0.600 −0.025, 0.015, −1.65, 0.110 0.092, 0.050, 1.82, 0.080 −1.10, 0.66, −1.68, 0.103

**Table 5 ijerph-20-00213-t005:** Stepwise regression analysis model with acetate and lactate as the criterion variables (Model 2).

	Acetate	Lactate
Adjusted *R*-squared	0.751	0.920
Predictors	β, SE, t, *p*	β, SE, t, *p*
Constant Age (years) Sex (male/female) BMI (%) Concentration of urinary pesticide markers DAP DMAP DEAP 3PBA Glyphosate Food intake frequency (/week) Rice Bread Pasta Potato Fish Meat Milk Fermented milk Beans Fermented beans Root vegetable Vegetable (others) Konjac Mushroom Seaweed Coffee Tea Supplement (Y/N) Beer can (350 mL) (/day) Other drinking (times/week) Smoking (*n*/*p*/c) Natural birth (Y/N) Breast milk (Y/N)	37.47, 5.82, 6.44, <0.001 −23.98, 4.86, −4.93, 0.001 8.89, 3.48, 2.55, 0.020 9.44, 3.08, 3.06, 0.007 3.02, 1.04, 2.91, 0.009 6.82, 1.41, 4.85, <0.001 −13.56, 2.29, −5.92, <0.001	−0.010, 0.447, −0.02, 0.981 −2.616, 0.493, −5.30, <0.001 0.876, 0.102, 8.61, <0.001 1.397, 0.154, 9.10, <0.001 −0.192, 0.156, −1.23, 0.239 −1.645, 0.188, −8.74, <0.001 0.025, 0.108, 0.23, 0.821 −0.218, 0.108, −2.02, 0.063

## Data Availability

FASTQ files of gut microbiota in the present study were previously deposited in the DNA Data Bank of Japan (DDBJ) under the accession number DRA009229 (Nishiwaki, Mov DIsord 2020). In the deposited dataset, sample names starting with “C” represent control subjects at Nagoya University and those starting with “F-C” represent control subjects at Fukuoka University. Metabolomic data are available upon request.

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
