# Peer review of "Effects of Pesticide Intake on Gut Microbiota and Metabolites in Healthy Adults"

_ijerph, 2022, doi:10.3390/ijerph20010213_

Round 1

Reviewer 1 Report (Previous Reviewer 2)

After first revision, the paper titled “Effects of pesticide intake on gut microbiota and metabolites in healthy adults” has significantly improved and most of the issues raised by the reviewers have been adequately addressed. This paper can be of good interest for the readers of International Journal of Environmental Research and Public Health and can be suitable for publication.

Author Response

Thank you very much for your kind advises you gave us last time. Please let me express my sincere gratitude once again. 

Reviewer 2 Report (Previous Reviewer 3)

The manuscript was successfully revised. Facts and conclusions drawn from the facts are now much clearer than before.

There are only two little points left:

Introduction, 1st paragraph, last sentence: This sentence would make more sense if it followed the very 1st sentence of the paragraph.

Introduction, lines 44 and 45: There seems to be a mistake. Shouldn't it read ... between organophosphorus insecticide (OP) exposure and ...

Author Response

Reviewer 2

The manuscript was successfully revised. Facts and conclusions drawn from the facts are now much clearer than before. There are only two little points left

Thank you VERY much for the useful advice. I sincerely appreciate your time and consideration.

Introduction, 1st paragraph, last sentence: This sentence would make more sense if it followed the very 1st sentence of the paragraph.

Response: This improvement will make it easier for readers to read. According to your advice, The last sentence has been moved to the top of the paragraph.

Introduction, lines 44 and 45: There seems to be a mistake. Shouldn't it read ... between organophosphorus insecticide (OP) exposure and ...

Response: The sentence "environmental pesticide" has been removed, as you pointed out.

This manuscript is a resubmission of an earlier submission. The following is a list of the peer review reports and author responses from that submission.

Round 1

Reviewer 1 Report

The authors assessed the relationship between pesticide exposure and fecal SCFAs and polyamines in the subjects preliminary. This is an interesting observation the gut microbiota and pesticide exposure in health Japanese subjects. I think however that there are a few improvements that should be made before publication.

 >Major Comments are as follows.

The authors described the subjects are adults of the group with an average age of 60+. However, does not the older age of the subjects have any effect on the gut microbiota and metabolites?

Another author described biological factors affecting the variability of the gut microbiota (e.g. Kandel Gambarte et al., 2022). Please describe further discussion of how age and other biological factors may affect gut bacteria.

>Here are the minor comments

 Line 260---For the vertical axis in Figure 1, isn't the vertical axis in the fourth figure from the top not “Varate” but “valeric acid”? For the vertical axis in the eighth figure from the top, full spelling of "PUT" is not shown in this manuscript.

Reviewer 2 Report

In this study, the authors explored the relationship between pesticide exposure and fecal SCFAs and polyamines in Japanese adults. The research population selected for this study was aged 69 ± 10 years. However, the obtained results were suggested to reveal the relationship between pesticide exposure and children’s health in the future. It was somewhat confused. Overall, this paper is not suitable for International Journal of Environmental Research and Public Health. Some comments are as follows:

1.     Line 37-39: The categories that were defined here were not comprehensive.

2.     Line 47-48: Different kinds of pesticides may have different toxicology mechanisms, and you must focus on a specific pesticide here. Therefore, this sentence is not appropriate.

3.     The Introduction was not well organized. Some of the important aspects for this study were not mentioned. For example, which kind of pesticide do you want to study? Why did you choose the elder adults as your research objects?

4.     Line 216-221: Just as you mentioned here, these people have different health conditions, and how can you guarantee the accuracy and representativeness of the following results?

5.     Line 288-290: The sample amount was not enough, and the selected population was somewhat not representative. Therefore, the obtained results can not conclude to extend to the study of children.

Reviewer 3 Report

One of my main concerns is about the data presented in fig. 1:

1. High and low OP-exposure: This was derived from urinary OP metabolite levels only?

2. Which samples were taken for the outlier box plots: simply the sample of the lowest and highest urinary OP metabolites, resp.? Or was a limit set between lower and higher levels and those samples are all involved?

3. It is not clear, whether the range of SFCA you present in table 2, column 5, is significantly far from the normal range (which should be the fact to really judge the findings of your study). You only compare the data within the small group of your participants in the study.

4. You present results for polyamines besides SFCA but there are no comments on these.

5. You cite a number of animal studies where effects of OP exposure on SFCA and others were presented but your study did not really reproduce those findings. So you explain this non-satisfactory fact with different exposure levels. But, please additionally keep in mind that a rat or a mouse is not human... So, maybe, the hypothesis you wanted to confirm, was not applicable to humans.

6. In lines 295-296 you give a correlation between fecal acetate level and attenuation of the microbiome. In lines 302-304 you state that there was no alteration of the microbiome. This is somewhat contradictory and does not increase the trust in your results.

7. In lines 321-322 you write that there was no association found between food intake and OP exposure markers. So there is another gap between hypothesis and results to make the latter reliable.

8. At last, I would like to say something positive: your analytical methods sound very sophisticated. We also do some methods with derivatization for GC-MS(/MS) (not OP metabolites which we do by LC-MS/MS only) but we never reached relative precesion lower than 10 % for derivatization methods.